# Potential Natural Products Regulation of Molecular Signaling Pathway in Dermal Papilla Stem Cells

**DOI:** 10.3390/molecules28145517

**Published:** 2023-07-19

**Authors:** Zar Chi Soe, Zin Zin Ei, Kittichate Visuttijai, Pithi Chanvorachote

**Affiliations:** 1Center of Excellence in Cancer Cell and Molecular Biology, Faculty of Pharmaceutical Sciences, Chulalongkorn University, Bangkok 10330, Thailand; zarchisoeygn96@gmail.com (Z.C.S.); hushushin@gmail.com (Z.Z.E.); 2Department of Pharmaceutics and Industrial Pharmacy, Faculty of Pharmaceutical Sciences, Chulalongkorn University, Bangkok 10330, Thailand; 3Department of Pharmacology and Physiology, Faculty of Pharmaceutical Sciences, Chulalongkorn University, Bangkok 10330, Thailand; 4Department of Laboratory Medicine, Institute of Biomedicine, University of Gothenburg, 405 30 Gothenburg, Sweden; kittichate.visuttijai@gu.se

**Keywords:** stem cells, hair follicles, dermal papilla cells, natural product-derived compounds, molecular pathways

## Abstract

Stem cells have demonstrated significant potential for tissue engineering and repair, anti-aging, and rejuvenation. Hair follicle stem cells can be found in the dermal papilla at the base of the follicle and the bulge region, and they have garnered increased attention because of their potential to regenerate hair as well as their application for tissue repair. In recent years, these cells have been shown to affect hair restoration and prevent hair loss. These stem cells are endowed with mesenchymal characteristics and exhibit self-renewal and can differentiate into diverse cell types. As research in this field continues, it is probable that insights regarding stem cell maintenance, as well as their self-renewal and differentiation abilities, will benefit the application of these cells. In addition, an in-depth discussion is required regarding the molecular basis of cellular signaling and the influence of nature-derived compounds in stimulating the stemness properties of dermal papilla stem cells. This review summarizes (i) the potential of the mesenchymal cells component of the hair follicle as a target for drug action; (ii) the molecular mechanism of dermal papilla stem cells for maintenance of their stem cell function; and (iii) the positive effects of the natural product compounds in stimulating stemness in dermal papilla stem cells. Together, these insights may help facilitate the development of novel effective hair loss prevention and treatment.

## 1. Introduction

Advancements in cell biology, stem cell research, and tissue engineering have not only provided new insights into the area of the cell biology’s control of hair cycling, hair regeneration, and hair loss but also have opened doors for developing novel tools for hair loss treatment. Currently, the most common drug treatments for hair loss are minoxidil and finasteride [1]. These drugs are effective in inducing new hair and preventing further hair loss but also have their limitations [2]. Hence, there is still a need for additional drug and compound treatments. Hair transplantation is another popular method for hair replacement [3]; however, this approach is limited by its cost and invasive protocols and requires drug and compound treatments to achieve optimal results [3]. Therefore, research on plant-derived compounds and extracts could contribute to new developments in this area. Studies have reported that natural product-derived compounds have beneficial effects on dermal papilla (DP) cells, which play a key role in controlling the elongated anagen phase and inducing hair growth [4].

Researchers have discovered the role of the cellular component of hair follicles (HF) in regulating hair growth, maintenance, and cycling and have identified the stem cell of the HF as an important player. Dermal papilla are stem cells that have a distinct category of cells situated at the base of the hair follicles, which are minute structures present in the dermal papilla and crucially involved in the process of hair growth. Moreover, the hair follicle stem cells have a pivotal role in the regular cycle of hair growth, regenerating and repairing hair follicles [5]. When there is injury or hair loss, these stem cells can be stimulated and undergo a process of rapid growth to generate new hair follicles [6]. Dermal papilla cells, also referred to as hair follicle cells, possess stemness characteristics that contribute to the growth and upkeep of hair. In addition, these cells exhibit stemness qualities, allowing them to self-renew and differentiate, which play crucial roles in maintaining hair and promoting its growth [7].

This understanding prompts the investigation of potential drug targets for developing effective treatments for hair loss. DP cells are specialized mesenchymal stem cells located at the base of HFs [8]. DP cells regulate hair growth by communicating with the surrounding cells via the release of growth factors and cytokines. Plant-derived compounds and extracts can enhance the activity of DP cells, leading to increased hair growth. The stem cell properties of the DP cells are recognized as essential for the hair-inducing effect of these stem cells [9]. DP cells can differentiate into multiple cell types, including skin cells and adipocytes, and the decrease in their pluripotency causes the decline of hair-regenerating properties [10]. In addition, hair loss, or alopecia, can occur when DP stem cells become damaged, depleted, or inactive. In androgenetic alopecia, the most common form of hair loss, DP cells become less active, resulting in shorter and thinner hair growth. HFs eventually become miniaturized and stop producing hair altogether. Stem cell therapies as well as other means to improve the stem cell properties of DP cells show promise for treating hair loss [11]. However, more research is needed to fully understand the efficacy and safety of these treatments.

Thus, advancements in cell biology and stem cell research have provided novel insights and tools for hair growth stimulation and hair loss prevention. Plant-derived compounds and extracts have shown promising results in promoting hair growth by enhancing the activity and stemness of DP cells. Further research in this field may enable the development of more effective and natural treatments for hair loss.

### 1.1. Stem Cells in the Hair Follicle

An HF is a mini organ that comprises several layers and types of cells that produce the hair shaft. An HF contains mesenchymal stem cells that are important for HF morphology and generation [12]. The interaction between these DP stem cells (DPSCs) and epithelial cells terms is termed epithelial–mesenchymal interaction (EMI) and is important for the maintenance of hair, regeneration of HFs, and hair cycling [13]. A healthy HF proceeds through the hair cycle including the anagen, catagen, and telogen stages, and hair disorders are known to be related to an abnormal hair cycle [14]. Advances in cell biology and stem cell research have highlighted the application of potential cells isolated from HFs, dermal papilla (DP) cells or hair follicle stem cells (HFSCs) [15]. HFs contains stem cells that originated during the embryonic development process, including from melanocyte, epithelial, and mesenchymal stem cells (MSCs) [16]. DPSCs maintain their stemness and stem cell properties including self-renewal and cell differentiation into different cell lineages. The stemness of the DPSCs was shown to be critical for hair growth, cycling, and regeneration [17]. The anagen growth phase of the hair cycle requires a close interaction between the DP and the HFSCs. In this phase, DP cells stimulate the functions of DPSCs and HF epithelial cells via DP-derived cytokines and growth factors [18]. During the regression phases including the catagen and telogen phases, the DP starts to detach from the HF, and the lack of a DP-derived signal causes the diminishing of the HF.

DP cells are specialized mesenchymal cells in HFs and play an important regulatory role in hair growth and turnover [17]. Studies have demonstrated that DP cells regulate hair growth, development, and cycling [19]. In addition, the size of the DP cell cluster at the base of the HF is directly proportional to the diameter of the hair shaft. The decrease in DP cell numbers can cause several hair loss conditions [20]. In addition, DP cells are well-established tools for investigating potential substances for anti-hair loss therapy [21]. Even though evidence indicates the potential of DP cells in the induction of new HF generation, isolated human DP cells cannot generate a new HF after being directly transplanted [13]. The loss of the ability of DP cells to form spheroids as well as the depletion of the DP stemness could be the rationale underlying the failure of the new HF generation.

Another stem cell population found in HFs are DPSCs. The progress of cell biology has helped identify and isolate this unique cell population located at the bulge region near the opening of the sebaceous gland [22]. DPSCs are slowly proliferating cells with quiescent characteristics. However, their activities are significantly increased during the telogen phase of the hair cycle [23]. Signaling and environmental factors are known to regulate the activation and condition of DPSCs. These stem cells receive signals from nearby cell populations including DP cells, adipose tissue, and immune cells. Two key signals known to control DPSCs’ function and activating status are the Wnt/β-catenin and bone morphogenetic protein (BMP) pathways [24]. The activated BMP signals suppress the activity of DPSCs, whereas the Wnt signals play the opposite roles. In addition, a high level of Wnt signaling is related to the growth of hair, as well as to HF formation [25]. Overall, the factors or compounds targeting the balance between Wnt and BMP signaling in DPSCs may serve as a novel strategy for the prevention of hair loss.

HF regeneration and hair maintenance requires good coordination between stem cell populations in the HF. DP cells secrete several growth factors and cytokines that regulate functions in other cells such as epithelial cells, melanocytes, and DPSCs. Signaling cytokines such as TGF-β2 and FGF-7 secreted by DP cells can activate DPSCs and drive DPSCs to function in the process of the new hair cycle via Wnt activation [26]. After anagen entry, DP function and the high cellular Wnt signaling are required to promote the progenitor cells to proliferate and produce the hair shaft [27]. Evidence suggests that the Wnt signal activity is essential for DP function during HF formation as well as anagen progression [28].

### 1.2. Indicators for Stemness Properties

Evidence has indicated that the expression of pluripotent transcription factors, including Oct4, Nanog, and Sox2, could be markers indicating the stem cell properties of mesenchymal cells. Through the process of induced pluripotent stem cell generation, the forced expression of Oct4, c-Myc, Sox2, and Klf4, or Oct4, Nanog, Sox2, and Lin28 could shift the cell phenotype and differentiation potential from that of a somatic cell to that of a pluripotent stem cell [29]. This discovery led to the concept that the cellular level of such transcription factors may be a critical marker for stemness. Consistent with many studies, the pluripotency transcription factors of Oct4, Nanog, and Sox2 are predominantly found in pluripotent stem cells, and their levels are found to be decreased during the differentiation process [30]. However, evidence has indicated that Oct4 and Nanog but not Sox2 are critical for the self-renewal potential in human embryonic stem cells [31]. Further experiments in mesenchymal cells demonstrated the high expression level of Oct4, Nanog, and Sox2 in mesenchymal cells (MSCs) [32]. In addition, the modification of their culture environment as well as the addition of specific substances could enable altered expression of these transcription factors and their stem cell properties as well as their involvement in the regulation of stem cell properties. Culturing the human mesenchymal stem cell in normal oxygen conditions is recognized to reduce their stemness, whereas hypoxic conditions could maintain such differentiation potentials [33]. Furthermore, cells from the early passages of the primary mesenchymal cells exhibit higher expression levels of Oct4 and Nanog. The expression levels of Oct4 and Nanog were found to gradually decrease during the culturing, and the late-passage cells exhibited a low level of both transcription factors, indicating the possible process of spontaneous differentiation, quiescence, or senescence. One study showed that the stemness of mesenchymal cells should at least in part be mediated by Oct4 and Nanog, as these two pluripotent transcription factors were augmented in mesenchymal cells cultured in hypoxic conditions. Under hypoxic status, these cells had a high differentiation potential. Importantly, forced knockdown of Oct4 and Nanog in such hypoxic stem cells significantly suppressed the proliferation rate, as well as their differentiation potential [34]. In addition, the study showed that Oct4 directly affected the stemness of HF stem cells by decreasing the level of cellular p21 [35]. Therefore, intervention, as well as the regulation of the environment, could maintain the expression of Oct4 and Nanog at a high level and could be an important means for maintaining the mesenchymal stemness. This concept is related to the fact that the widely used drug for treating hair loss and androgenetic alopecia, i.e., finasteride, has the potential to stimulate stemness in DP cells. Finasteride can enhance the expression of pluripotent transcription factors Sox-2 and Nanog in human DP cells. In addition, the widely used stem cell marker CD133 was demonstrated as a biomarker able to indicate the stemness of DP cells [36]. The expression of CD133 was shown to reflect the HF-inducing effect of DP cells. Mechanistically, CD133-positive DP cells could secrete the Wnt ligand that facilitates the EMI and enhance the activity and growth of HFs [37].

## 2. Molecular Pathways Regulating Stem Cell Properties of Hair

Having shown that stem cells are the key regulators affecting hair growth and loss, a better understanding of the mechanisms that regulate stem cell properties could help pave the way toward inventing effective therapeutic approaches. Several stemness-stimulating signals have been identified, and several of them are known for regulating hair stem cells. When these pathways are disrupted, stem cells fail to self-renew, causing the depletion of the stem cell pool and a loss of regeneration potential [38].

### 2.1. Wnt/β-Catenin Pathway

The Wnt signaling pathway consists of canonical and noncanonical signaling pathways. The noncanonical signaling pathway consists of the cell polarity signaling pathway and the Wnt/Ca^2+^ pathway. However, the primary mechanism for DP cell proliferation and differentiation is via the canonical Wnt signaling.

The Wnt/β-catenin signaling pathway is critical in maintaining the hair-inducing activity of DP cells. Wnts 2, 7b, 10a, and 10b are involved in HF formation [39]. At early phases of HF morphogenetic development, the expression of Wnt 10b, 10a, and 5a is selectively increased.

In the absence of Wnt, the phosphorylated catenin is degraded by the destruction complex of GSK-3β (glycogen synthase kinase-3 β), APC (adenomatous polyposis coli), Axin (axis inhibition protein), and CK1α (casein kinase 1 α). This complex is subsequently degraded by the proteasome via β-TrCP/SKP (β-transducin repeat-containing protein/s-phase kinase-associated protein), the ubiquitin-proteasomal ligase complex.

The canonical Wnt pathway starts with the interaction of the frizzled receptor (FZDR) with its ligands to mediate signal transduction. The FZDR and a co-receptor of the low-density lipoprotein-related protein (LRP) interact with secreted or extracellular Wnt in the canonical Wnt pathway. This association induces signaling that inhibits glycogen synthase kinase-3β (GSK-3β). The inhibition of GSK-3β stabilizes β-catenin because GSK-3β is the key kinase enzyme that phosphorylates β-catenin for ubiquitin-proteasomal destruction [40]. The TCF/lymphoid enhancer factor (LEF) binds to the stabilized β-catenin and translocates to the nucleus. The complex of β-catenin-TCF/LEF initiates the transcriptional activation of genes that regulate cell proliferation and stemness [39] (Figure 1).

### 2.2. Notch Signaling Pathway

For HFs to grow and mature, the Notch signaling pathway is necessary in several steps. Importantly, Notch signaling controls the interfollicular epidermis lineage or the HF commitment process during epidermal development. The Notch receptor protein family is a single type-I transmembrane receptor. In mammals, there are four receptors (Notch 1-4) and five matching ligands (Delta-like 1 (Dll-1, 3, 4,), Jagged-1, and Jagged-2) that bind via cell-to-cell contact [41] (Figure 1). Delta-like 1 is expressed in the embryonic HF DP region and bulb but not in mature HFs. Delta-1 is predominantly expressed in mesenchymal cells of the presumed dermal papilla from the preliminary follicle development. Notch-1 is present in abundance in the follicular bulb of neonatal vibrissa follicles but is significantly reduced or missing in the bulb cells close to the DP. SOX2/CD133+ DP cells can be activated by the Notch signaling pathway, and their stem cell properties are maintained by Shh signaling [42]. Consequently, Notch signaling may control Shh signaling to maintain the heterogeneity of DP cells. Androgenetic alopecia has been shown to be involved with the Notch signaling system.

Driskell et al. previously demonstrated that the Notch, Shh, IGF, and integrin pathways were preferentially elevated in the DP cells of zigzag-type HFs, whereas the Wnt, FGF, and BMP pathways were overwhelmingly expressed in guard/awl/auchene follicle DP cells [43]. This finding suggests that the signaling pathways play a lineage-specific role. Furthermore, co-expression of Notch-1 and its ligand, Jagged-1, was discovered in the upper follicle bulb and was likely correlated with DP cell development.

### 2.3. TGF-β/BMP Signaling Pathway

The bone morphogenetic protein (BMP) of the transforming growth factor-β (TGF-β) family plays an important role in the initiation of HF growth. Among the many types of BMP, BMP2 and BMP4 have been shown to be involved in HFs. BMP2 is dominantly expressed in the growth period of HF and its level declines when the HF goes to regression. The expression of BMP4 is low at the growth phase and mainly upregulated during the regressive period of the hair cycle. In addition, BMP4 and BMP6 arrest follicle proliferation during the telogen phase [44].

BMP signaling is also required by DP cells to initiate HF growth and governs the progression of the transit-amplifying lineages formed from DPSCs via Smad 1/5/8 target genes. TGF-β ligands bind to the receptors of the cell membrane and phosphorylate intracellular Smad 2,3 while BMP ligands phosphorylate Smad 1,5,8. Active Smads associate with co-Smad4 and translocate to the nucleus. TGF-β signaling, however, has been directly linked to the induction and advancement of the regression phase of the HF cycle (catagen phase). Furthermore, TGF-β signaling can antagonize the BMP-mediated suppression of HF stem cell activation [45]. TGF-β binds to the type-II receptor, whereas BMP activates the type-I receptor, resulting in a binary complex that creates the downstream complexes that regulate target gene expression (Figure 1).

### 2.4. Sonic Hedgehog Shh Pathways

Sonic hedgehog (Shh) signaling regulates epithelial growth in HFs and controls the expression of a subset of DP-specific signature genes. During the maturation phase of the HF, the depletion of Shh causes a decrease in the HF keratinocyte proliferation rate. In Shh signaling, GLI2 is identified as the major transducer, while GLI1, whose expression is transcriptionally controlled by GLI2, is thought to be a secondary function in potentiating the Shh response. Importantly, Shh signaling is predominantly important for the DP cell function. Shh can promote HF neogenesis in scarred wounds by generating DP cells [46] (Figure 1).

Conversely, the development of HFs also depends on the negative regulation of Shh signaling. PTCH1 and 2 are recognized inhibitors of Shh signaling, and Ptch1 is a GLI target gene. PTCH1- and 2-mediated Shh signaling inhibition is critical for controlling epidermal differentiation [47]. Many studies have found an increase in the number of immature HFs in the induction phase in Shh/Gli3/skin, where the repressor form of GLI3 (GLI3R), another negative regulator of Shh signaling, is missing. Furthermore, keratinocyte differentiation fails in Shh/Gli3/skin, causing the failure of HF maturation. The ability of HH signaling to maintain progenitor cells is critical for both epithelial and mesenchymal cell fate determination during HF development.

### 2.5. PI3K/AKT Signaling Pathway

The phosphoinositide 3-kinase, PI3K/protein kinase B (Akt) pathway is important in controlling the proliferation and survival of many cells. Akt signaling is also recognized as a regulator of stemness and the maintenance of stem cell properties. This effect of the signal has been linked to the induction of the Wnt/β-catenin pathway.

The activation of PI3K/Akt could inhibit the function of GSK-3β, a negative modulator of the Wnt/β-catenin pathway. Akt is a serine/threonine kinase that is a direct downstream target of PI3K and is essential for cell proliferation, survival, and apoptosis [48]. HF stimulation of the PI3K/Akt pathway is considered as a promising therapeutic target in hair regeneration based on evidence of PI3K/Akt action in HF regeneration.

PI3K can be activated when a relevant ligand attaches to the receptor. PI3K activation results in the formation of the second messenger phosphatidylinositol 3,4,5-triphosphate (PIP3) on the plasma membrane. PIP3 interacts with PI3K-dependent kinase 1 (PDK1), which can phosphorylate Thr308 on the Akt protein, activating Akt. PI3K-dependent kinase 2 (PDK2) can phosphorylate Ser473 on the Akt protein, activating it. Activated Akt then has a significant impact on downstream target proteins by activating or inhibiting associated downstream molecules such as the mechanistic target of rapamycin complex (mTORC), GSK3, and Forkhead box O (FOXO) [25] (Figure 1).

## 3. Effects of Natural Product-Derived Compounds and Extracts That Stimulate Hair Follicle Stem Cells

Many studies support the use of natural products to manage hair loss and develop regeneration strategies. Active drugs that produce or improve stem cell signaling and the properties of DP cells may assist in the therapeutic treatment of hair loss. The activation of the stem cell-related transcription factors was found to be connected to an increase in the Wnt β-catenin signal in DP cells treated with finasteride [49]. The benefits of plant-derived substances and extracts on DP cells have been demonstrated by many studies (Table 1).

### 3.1. Geranium Sibiricum (L.) Extract

*Geranium sibiricum* (L.) extract (GSE), which has been used as a medicinal plant extract, that possesses significant quantities of polyphenolic compounds, including corilagin and gallic acid (Figure 2). It can enhance hair growth in DP cells by inducing the cellular hepatocyte growth factor (HGF) and vascular endothelial growth factor (VEGF). Boisvert et al. confirmed that GSE therapy dramatically increased hair growth in vitro and in vivo by increasing DP proliferation and migration. GSE treatment also altered the expression of HGF, VEGF, and TGF-1, all of which are involved in the growth and inhibition of hair follicular cells. TGF-β1 function is related to hair loss, and the overexpression of TGF-β1 in hair growth inhibits cell growth by accelerating the transition between catagen and telogen. By preserving the catagen phase of HFs and the survival of follicular cells, Boisvert et al. reported that TGF-β1 expression was downregulated in both in vitro (DP cells) and in vivo models of GSE extract treatment and that hair growth was stimulated [50].

### 3.2. Ishige Sinicola

Ishige sinicola is a brown algae with various bioactivities, including osteoblastic bone formation and anti-inflammatory properties [66]. Octaphlorethol A is a constituent extracted from I. Sinicola (Figure 3). *I. sinicola* therapy for three weeks lengthened hair fibers and induced the hair shaft anagen phase in rat vibrissa follicles *I. sinicola* therapy boosted β-catenin protein levels and GSK-3β phosphorylation in cultured DP cells. By upregulating the expression of cyclin E, CDK2, β-catenin, and p-GSK3 and downregulating that of p27kip1, *I. sinicola* extract promoted the proliferation of DP cells. However, the well-known *I. sinicola* chemical octaphlorethol A may promote the growth of DP cells and suppress the activity of 5-reductase. Kang et al. indicated that in treating alopecia, octaphlorethol A can demonstrate the therapeutic effects of both minoxidil and finasteride. Their results proved that *I. sinicola* extract increased the initiation of anagen in C57BL/6 mice in a dose-dependent manner and that octaphlorethol A may stimulate hair growth by increasing the number of DP cells in a manner similar to minoxidil treatment. Therefore, *I. sinicola* extract may aid in treating alopecia by increasing DP cell proliferation then activating the β-catenin pathway and inhibiting 5-reductase [51].

### 3.3. Resveratrol

Resveratrol (Figure 4) was discovered as a plant polyphenol in the root of *Veratrum grandiflorum*. The pharmacological benefits of resveratrol include antitumor, antioxidant, anti-inflammatory, and antiaging properties. Additionally, resveratrol treatment can hasten the healing of wounds, increase the quantity of HFs in the skin of young mice, and encourage the growth of HFs. Resveratrol-containing polyphenols also controlled the expression of IGF-1 and KGF, which promote the β-catenin pathway, and TGFβ1 expression, which is critical in preserving the niche of HFSCs and is assumed to perform roles in encouraging hair development [52]. Furthermore, Zhang et al. discovered that combining resveratrol with pyridine 2,4 dicarboxylic acid diethyl ester shields hair matrix cells in HFs from the harmful effects of oxidative stress and improves human hair thickness [53,67]. Consequently, it is thought that resveratrol can increase hair development and may be a candidate medication to treat hair loss, although further research is required.

### 3.4. Polygonum Multiflorum

The extract of *Polygonum multiflorum* mainly containing 2,3,5,4′-tetrahydroxystilbene-2-O-β-D-glucoside (TSG, another resveratrol glycoside) (Figure 5) stimulated hair growth by increasing Shh expression. Several reports have demonstrated the hair growth effects of *P. multiflorum* extract. A histological examination of C57BL/6 mice revealed that both topical and oral administrations of *P. multiflorum* extract increased the size and number of HFs by upregulating β-catenin and Shh expression [54]. Li et al. assessed the hair growth activity and possible mechanisms of *P. multiflorum* Radix. FGF-7, Shh, β-catenin, IGF-1, and HGF expression levels were measured from the 3rd to 6th week in mice, and they found that both the oral and topical administration of *P. multiflorum* Radix could promote hair growth. The hair growth promotion effect of oral *P. multiflorum* Radix was most probably mediated by the expression of FGF-7, whereas topically administrated *P. multiflorum* Radix promoted hair growth by the stimulation of Shh expression. This could enhance the application of natural products in the field of hair promotion in the future [55].

### 3.5. Miscanthus Sinensis var. Purpurascens

The flower extract of *Miscanthus sinensis* var. *purpurascens* contains active compound 4-Hydroxybenzaldehyde (Figure 6). It has been demonstrated to induce the stem cell properties of DP cells via upregulating the expression of β-catenin and HGF [56]. As previously stated, VEGF enhances HF growth, whereas TGF-β1 inhibits HF growth. TGF-β1 causes the catagen phase of the hair cycle and can cause morphological alterations and apoptosis in HFs. The vascularization of the hair DP can be facilitated by VEGF, which may help to extend the anagen phase [57]. Jeong et al. examined the expression of VEGF and TGF-β1 in mouse dorsal skin. When compared with the effect of minoxidil, the flower extract of *M. sinensis* var. *purpurascens* dramatically reduced the production of TGF-β1 both in vitro and in vivo. *M. sinensis* var. *purpurascens* flower extract may stimulate human hair growth by activating the ERK pathway and prolonging the anagen phase by boosting β-catenin levels in human DP cells (hDPs). Furthermore, the *M. sinensis var. purpurascens* flower extract may benefit stress-related hair loss by decreasing mast cell degranulation, substance P (SP), neuropeptides, and TGF-β1 expression [56].

### 3.6. Quercetin

Quercetin (Figure 7), a component of the *Hottuyunia cordata* extract, has also been found to promote hair development. Kim et al. conducted the study by using hDPs to assess the effects of the extract and discovered considerable changes in mitochondrial function. The mitochondrial membrane potentials and NADPH production were found to be raised, implying improved mitochondrial function. Bcl2, a marker for the anagen phase and an enhancer of cell survival, was also expressed more frequently. Additionally, expressions of VEGF, bFGF, and KGF growth factors had increased, and phosphorylation of Akt, ERK, and CREB as well as the cell proliferation marker Ki67 had also increased. The use of quercetin was found to activate the MAPK/CREB pathway, which enhanced the expression of growth factors and caused an increase in hair growth [58]. Wikramanayake et al. demonstrated improved hair growth in comparison with a placebo in skin lesions of mice after the mice were administered with quercetin. Consequently, quercetin may be a potential therapy choice for alopecia [68].

### 3.7. Centipeda minima (L.) A. Braun & Asch Extract

*Centipeda minima* (L.) A. Braun & Asch extract includes 12 primary chemical ingredients that have been identified, including flavonoids, polyphenolic acid, and sesquiterpene lactones such as Brevilin A (Figure 8). Kim et al. proved that *C. minima* (L.) enhanced hair development by inducing the proliferation of DP cells and increasing the expression of the FZDR (frizzled receptor), Wnt5a (Wnt family member 5a), and VEGF, indicating an increase in growth factor expression (VEGF and IGF-1). In DP cells, extracellular signal-regulated kinase (ERK) and c-Jun N-terminal kinase (JNK) were phosphorylated, and β-catenin had accumulated. Their findings revealed that *C. minima* (L.) promotes hair growth via Wnt/β-catenin (a protein involved in the development of HFs and sebaceous glands), ERK, and JNK signaling pathways. *C. minima* (L.) exhibited drug-likeness and bioavailability and could be used to control hair loss and induce hair growth [59].

### 3.8. Trapa Japonica

*Trapa japonica*, a floating aquatic plant found in ponds and other slow-moving bodies of water, is used in East Asia as both a traditional meal and a tonic [69]. The AC2 peptide was isolated from *Trapa japonica* (Figure 9). Nam et al. demonstrated that *T. japonica* fruit extract increased hair growth activity, cell cycle progression, and vascular expansion linked to VEGF in hDP cells. Their findings demonstrated that when hDP cells were treated with *T. japonica* fruit extract, the expression of p-Akt, p-ERK, and p-GSK-3 increased in a dose-dependent manner. These findings suggested that *T. japonica* fruit extract influenced hDP cell proliferation and cell cycle progression via the Akt/ERK/GSK-3 pathway, decreased type-I 5-reductase activity, and prevented apoptosis. IGF-1 is required for anagen phase growth and suppresses apoptosis during the catagen phase of the hair cycle, and KGF is most effective in stimulating hair growth factors such as VEGF and type-I 5-reductase inhibitors [1]. In this study, the authors also determined that IGF-1 and KGF are hair development biomarkers in the hair bulb. IGF-1 and KGF protein expression was validated using a three-dimensional cell culture model that matched the environment of actual HFs using human DP cells, confirming that *T. japonica* fruit extract stimulates human DP cells to increase hair growth and suppress hair loss [63].

### 3.9. Tocotrienols Rich Fraction (TRF)

Tocotrienols belong to the vitamin E family and are often found in plant-based sources including annatto, palm oil, rice bran, coconut oil, and barley. Annatto and palm oil are the richest natural sources of tocotrienols, with a tocotrienol content of approximately 100% and 70%, respectively. Tocotrienols are a group of fat-soluble antioxidants that have phenolic heads and are like tocopherols but with different side chains. Tocotrienols have an unsaturated isoprenoid side chain with three double bonds (phytyl chain), whereas tocopherols have a saturated side chain with no double bonds (Figure 10).

TRF has potential as a tool for promoting HF regeneration through a novel epidermal pathway. TRF decreased the expression of epidermal E-cadherin and translocated β-catenin into the nucleus. Nuclear β-catenin interacted with Tcf3 (transcription factor 3). The β-catenin-Tcf3 complex induced the expression of pluripotent factors such as Oct4, Sox9, Klf4, c-Myc, and Nanog. This novel pathway induces hair follicular regeneration in DP cells [60] (Figure 11).

### 3.10. 1α,25-dihydroxyvitamin D_3_ (VD_3_)

1α,25-dihydroxyvitamin D_3_ (VD_3_), also known as cholecalciferol, is a precursor to the biologically active form of vitamin D (Figure 12). After it is synthesized or ingested, VD_3_ is first converted into 25-hydroxyvitamin D_3_ (25(OH)_2_D_3_) in the liver. This form of vitamin D is then transported to the kidneys, where it is further converted into the biologically active form of vitamin D, known as 1,25-dihydroxyvitamin D_3_ (1,25(OH)_2_D_3_), by the enzyme 1α-hydroxylase.

The activation of transforming growth factor β2 (TGF-β2) promoted the catagen phase of the hair cycle and was mediated by the activation of Wnt10b, an alkaline phosphatase (ALP) in human DP cells, for its hair-inducing ability. Evidence suggests that vitamin D_3_ (VD_3_) can promote the functional differentiation of DP cells, which are the specialized cells that play a key role in HF formation and growth. This led to an interest in the potential use of VD_3_ as a tool for hair regeneration therapies [61].

### 3.11. Red Ginseng Oil

Red ginseng oil (RGO) is derived from the roots of the *Panax ginseng* Meyer plant. It is produced using organic solvent extraction or supercritical fluid extraction methods, commonly used to extract oil from plants. RGO contains a high level of unsaturated fatty acids such as oleic acid, linoleic acid, and linolenic acid (Figure 13) with lipophilic components such as phytosterols, tocopherols, and polyacetylenes. RGO promoted the premature anagen entry of telogenic HFs through the activation of the Wnt/β-catenin and Shh/Gli-1 signaling pathways in testosterone-treated C57BL/6 mouse models. Moreover, this pathway could significantly induce cyclins D1 and E during RGO treatment. RGO treatment can increase the levels of growth factors such as vascular endothelial growth factor (VEGF) and insulin-like growth factor 1 (IGF-1) for improved hair growth in C57BL/6 mouse models [62].

### 3.12. Salvia Plebeian

*Salvia plebeian* R. Brown (*SP*) (Labiatae) contains flavonoids, monoterpenoids, sesquiterpenoids, diterpenoids, triterpenes, and phenolic acids. Rosmarinic acid and homoplantaginin (Figure 14) are the primary components that make up the phenolic and flavonoid compounds found in SP. An *SP* extract activated the Wnt/β-catenin signaling pathway by increasing the expression of β-catenin and phosphorylated GSK3β. The *SP* extract had a positive effect on the survival and proliferation of human DP cells by increasing the expression of the Bcl-2/Bax ratio and activating cell proliferation-related proteins, ERK and Akt. The *SP* extract effect was achieved by downregulating the expression of TGF-β1, known to play a role in regulating hair growth, through suppression of the Smad 2/3 cascades. Therefore, *SP* can induce hair growth by preventing the transition from the anagen to catagen phase of the hair growth cycle in human DP cells [63].

### 3.13. Ginkgo Biloba Extract

*Ginkgo biloba* extracts contain flavonoids (flavonol glycosides), diterpene trilactones including ginkgolides A, B, and C, and pentanorditerpene bilobalide (Figure 15). The secretion of VEGF (vascular endothelial growth factor) is enhanced by *G. biloba* in DP cells for the growth of the cycling of HFs. At the molecular level, the Akt, ERK1/2 and β-catenin signaling pathways are promoted in HFs. Treatment with *G. biloba* extracts can induce an upregulation of the cytoplasmic level of p-GSK3β and a downregulation of Dkk1 (Wnt antagonist) in DP cells. The Wnt/β-catenin pathway, which is controlled by the Akt and ERK pathways, can be activated by treatment with *G. biloba* extracts [64].

### 3.14. Centella asiatica (L.) Extract

*Centella asiatica* (L.) extract contains phenols and flavonoids that possess high antioxidant activity. Asiaticosides (Figure 16), a type of saponin or triterpenoid, are the main active components found in C. asiatica (L.) extract, which confers the positive hair growth promotion effect in human follicle DP cells through the expression of VEGF to promote the regeneration of the hair [65].

## 4. Conclusions

Accordingly, our studies have identified natural products or compounds with potential as active molecular signaling agents for the treatment of hair regeneration. Overall, our study highlights the importance of investigating natural products as alternative treatments for hair regeneration, with the potential to minimize side effects. It also serves as a valuable resource for researchers and clinicians working in this field. Furthermore, these studies will play a significant role in the development of cosmetics or treatments that can effectively enhance the process of hair regeneration.

## Figures and Tables

**Figure 1 molecules-28-05517-f001:**
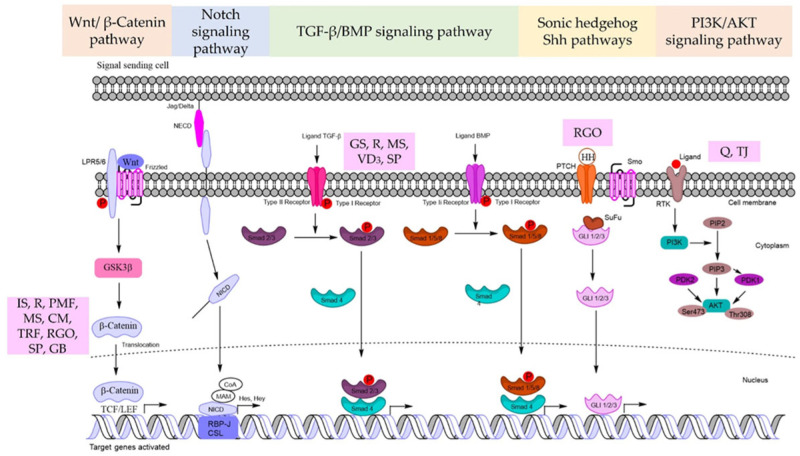
Summarized signaling pathways for natural product-derived compounds with hair regeneration in DP cells. GS = *Geranium sibiricum* (L.) extract, IS = *Ishige sinicola*, R = Resveratrol, PMF = *Polygonum multiflorum*, MS = *Miscanthus sinensis var. purpurascens*, Q = quercetin, CM = *Centipeda minima* (L.) A. Braun & Asch extract, TJ = *Trapa japonica*, TRF = tocotrienols rich fraction, VD3 = 1α,25-dihydroxyvitamin D3, RGO = red ginseng oil, SP = *Salvia plebeian*, GB = *Ginkgo biloba* extract.

**Figure 2 molecules-28-05517-f002:**
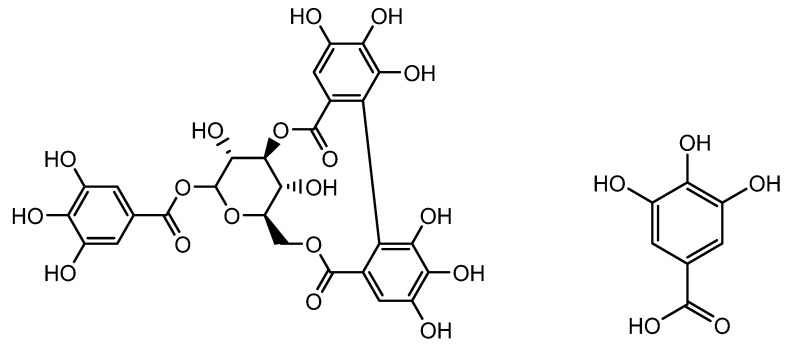
Structure of corilagin and gallic acid.

**Figure 3 molecules-28-05517-f003:**
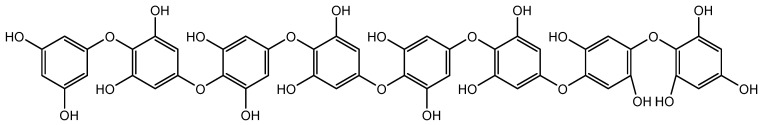
Structure of Octaphlorethol A.

**Figure 4 molecules-28-05517-f004:**
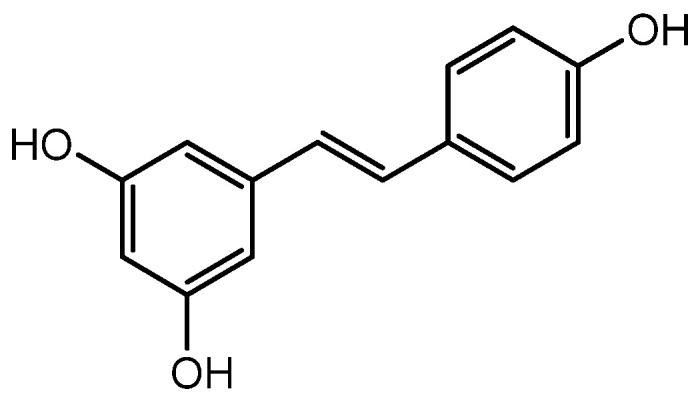
Structure of Resveratrol.

**Figure 5 molecules-28-05517-f005:**
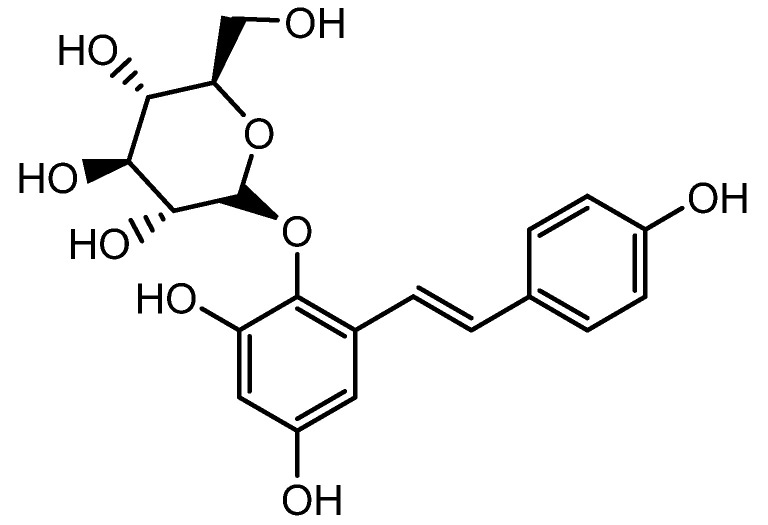
Structure of 2,3,5,4′-tetrahydroxystilbene-2-O-β-D-glucoside.

**Figure 6 molecules-28-05517-f006:**
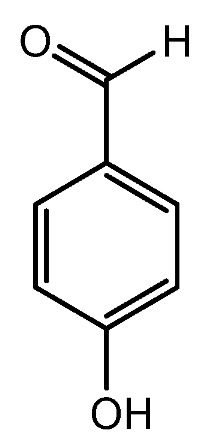
Structure of 4-Hydroxybenzaldehyde.

**Figure 7 molecules-28-05517-f007:**
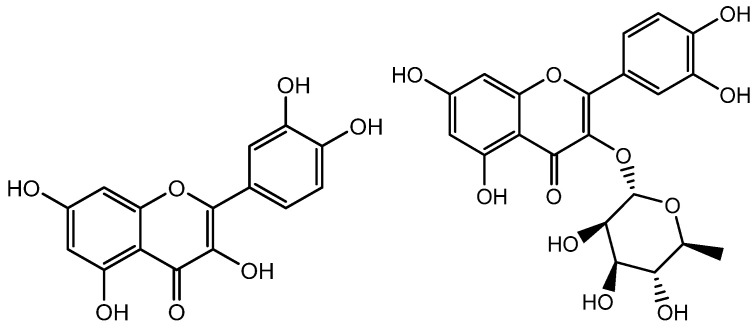
Structure of Quercetin and Quercitrin.

**Figure 8 molecules-28-05517-f008:**
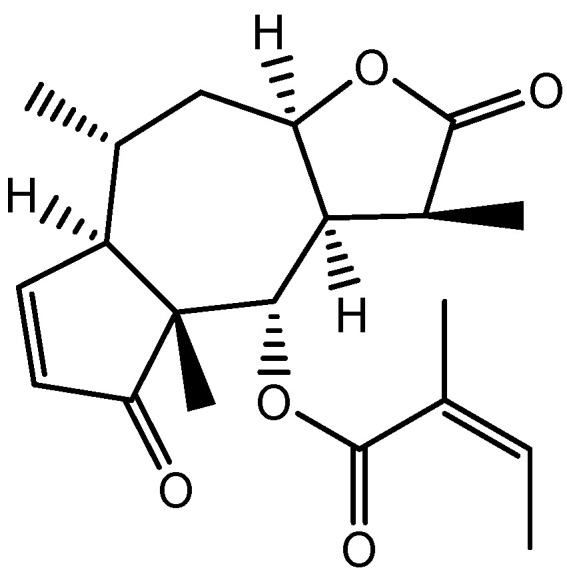
Structure of Brevilin A.

**Figure 9 molecules-28-05517-f009:**
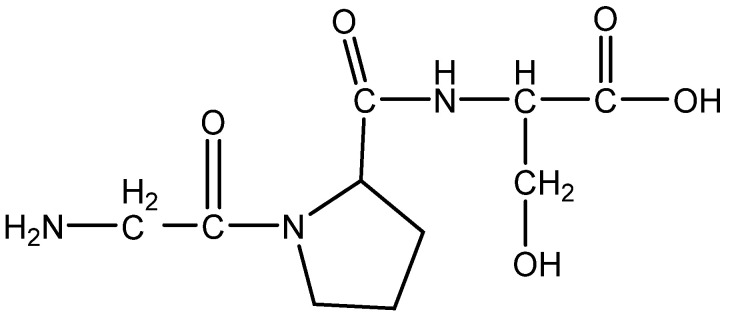
Structure of AC2 peptide.

**Figure 10 molecules-28-05517-f010:**
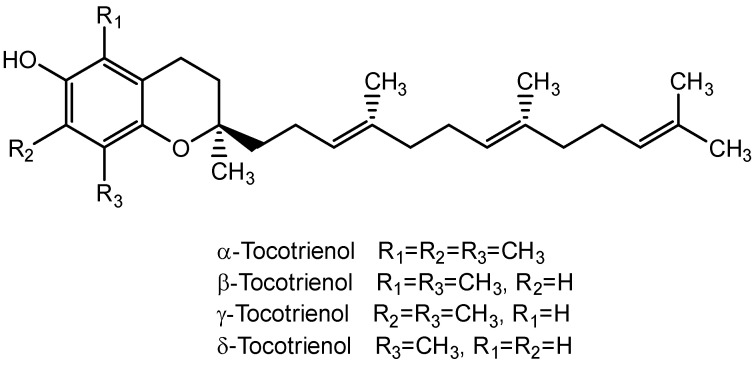
Structure for types of tocotrienol.

**Figure 11 molecules-28-05517-f011:**
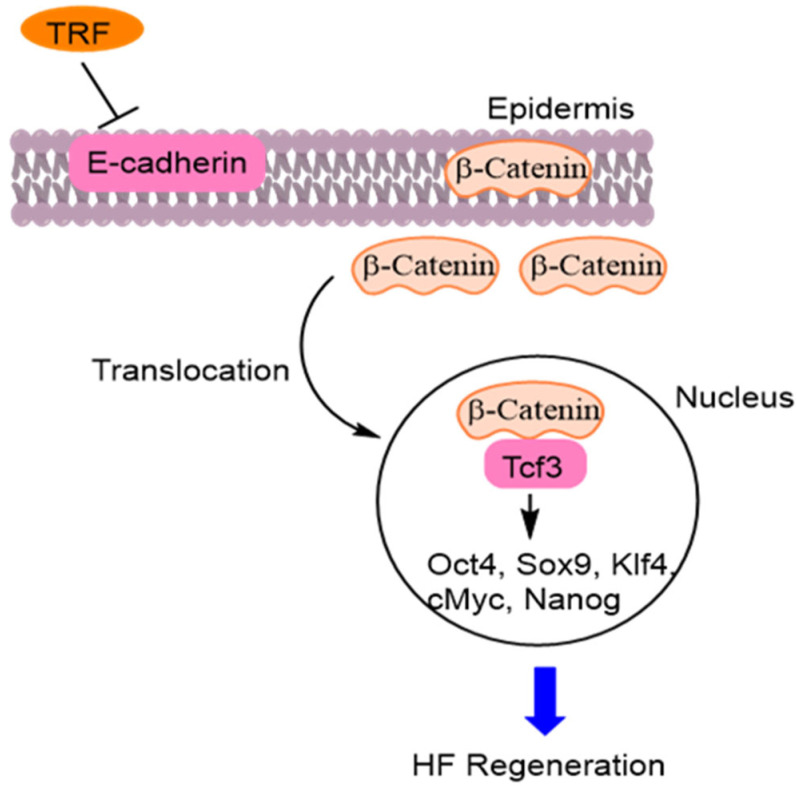
Tocotrienol rich fraction (TRF) decreased expression of epidermal E-cadherin and translocated β-catenin into the nucleus. Nuclear β-catenin interacted with Tcf3 (transcription factor 3). The β-catenin-Tcf3 induces pluripotent factors such as Octamer-binding transcription factor 4 (Oct4), SRY-Box Transcription factor 9 (Sox9), Krueppel-like factor 4 (Klf4), cellular myelocytomatosis oncogene (c-Myc), and Nanog [60].

**Figure 12 molecules-28-05517-f012:**
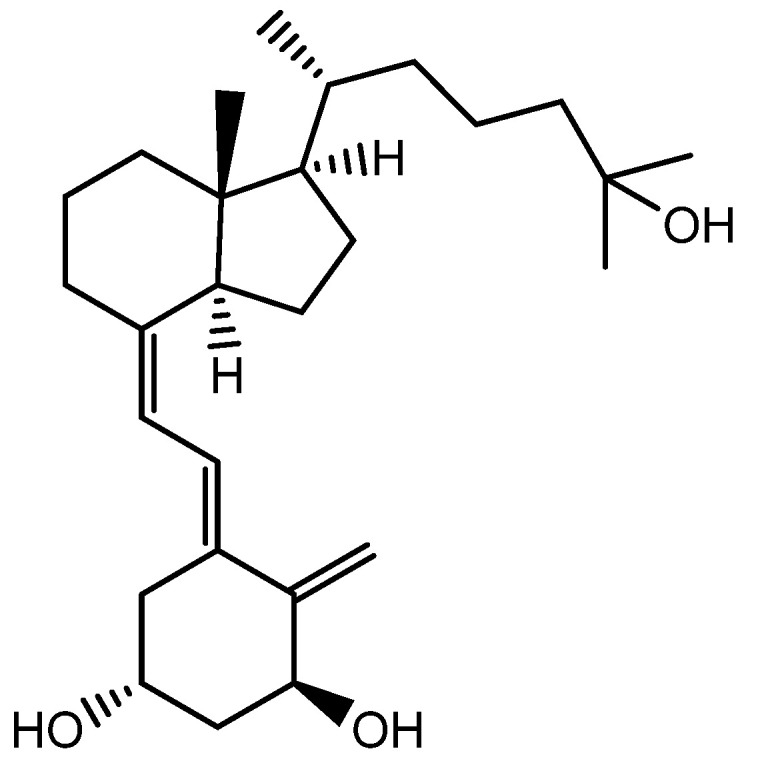
Structure of 1α,25-Dihydroxyvitamin D_3_.

**Figure 13 molecules-28-05517-f013:**
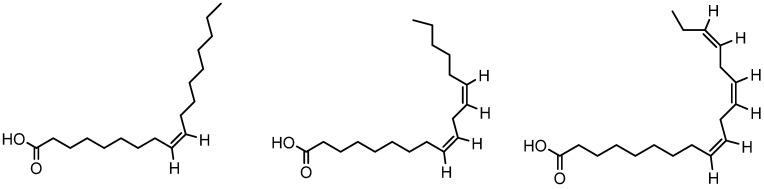
Structure of oleic acid, linoleic acid, linolenic acid.

**Figure 14 molecules-28-05517-f014:**
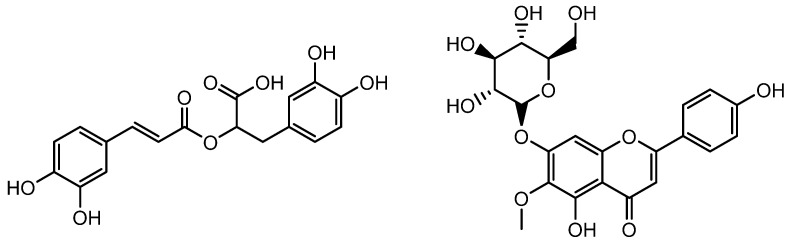
Structure of Rosmarinic acid and homoplantaginin.

**Figure 15 molecules-28-05517-f015:**
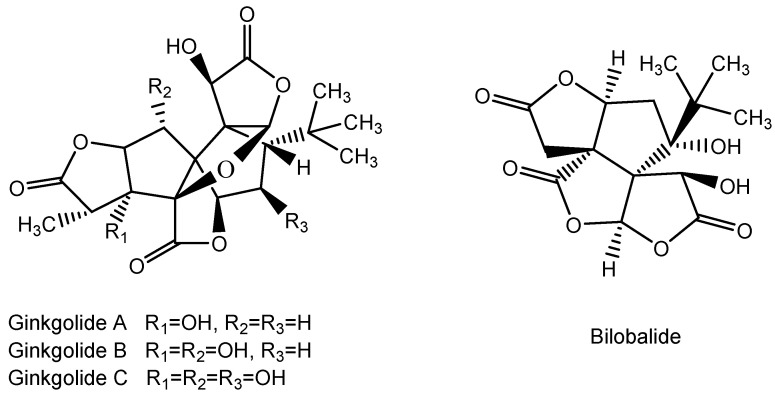
Structure of Ginkgolide A, B, and C and Bilobalide.

**Figure 16 molecules-28-05517-f016:**
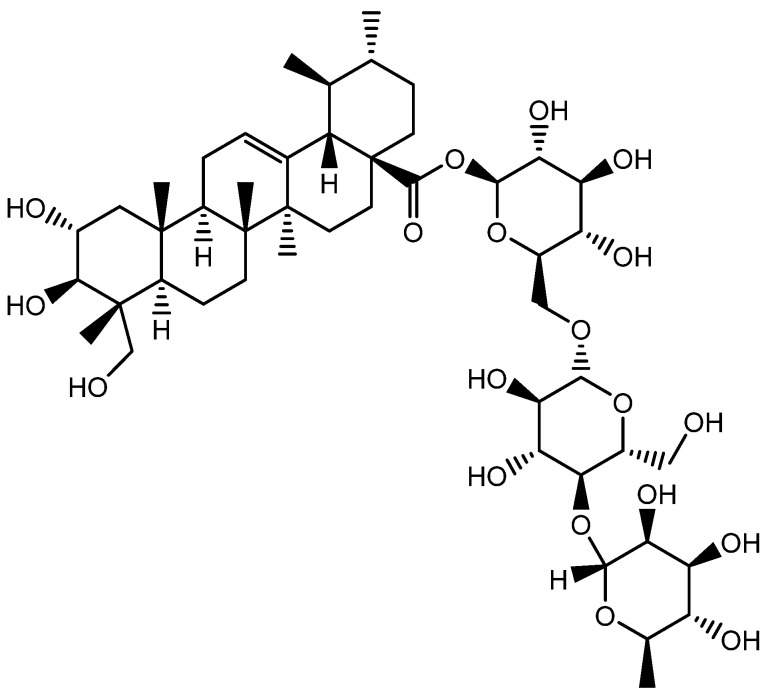
Structure of Asiaticoside.

**Table 1 molecules-28-05517-t001:** The representative natural product-derived compounds and extracts with hair regeneration.

Sources	Natural Product-Derived Compounds and Extracts	Chemical Compounds	Mechanism of Action	References
Minnesota wildflowers	*Geranium sibiricum* (L.) extract	Corilagin, gallic acid	Upregulation of VEGF and HGF Downregulation of TGF-β	[50]
Brown algae	*Ishige sinicola*	Octaphlorethol A	Progression of anagen Upregulation of Cyclin E, CDK2, β-catenin, and phospho-GSK3β Downregulation of p27^kip1^	[51]
Grapes, apples, blueberries	Resveratrol	3,5,4-trihydroxystilbene	Activation of β-catenin pathway and TGF-β1 gene expression Controlling IGF-1 and KGF gene expression	[52,53]
Roots of Polygonum multiflorum	*Polygonum multi florum*	2,3,5,4′-tetrahydroxystilbene-2-O-β-D-glucoside (THSG)	Upregulation of β-catenin and Shh	[54,55]
*Miscanthus Sinensis var Purpurascens* Grass	*Miscanthus sinensis var. purpurascens*	4-Hydroxybenzaldehyde	Activation of ERK, TGF-β1, HGF, and β-catenin Decreasing mast cell degranulation, substance P(SP), and neuropeptides	[56,57]
Onions, grapes, berries, cherries, broccoli, citrus fruits	Quercetin	Quercitrin	Activation of MAPK/CREB Increasing the expression of VEGF, bFGF, KGF, Akt, Erk, and CREB	[58]
Spreading sneeze weed	*Centipeda minima* (L.) A. Braun & Asch extract	Brevilin A	Increasing the expression of FZDR, Wnt5a, and VEGF Activation Wnt/β-catenin, ERK, and JNK	[59]
Water chestnut	*Trapa japonica*	AC2 peptide	Increasing the expression of p-Akt, p-ERK, and p-GSK-3	[1]
Palm oil	Tocotrienols rich fraction (TRF)	α, β, γ, δ -tocotrienol	Decreasing the expression of E-cadherin Increasing β-catenin Increasing the expression of pluripotent factors such as Oct4, Nanog, etc.	[60]
Milk, cheese	1α,25-dihydroxyvitamin D_3_ (VD_3_)	1α,25-dihydroxyvitamin D3	Activation of TGF-β2 Activation of Wnt10b	[61]
*Panax ginseng* Meyer	Red Ginseng Oil	Oleic acid, linoleic acid, linolenic acid	Activation of Wnt/β-catenin and Shh/Gli-1 Expression of VEGF	[62]
*Salvia* plant	*Salvia plebeian*	Rosmarinic acid, Homoplantaginin	Activation of Wnt/β-catenin pathway Downregulation of TGF-β1 Activation of Akt pathway	[63]
*Ginkgo biloba*	*Ginkgo biloba* extract	Ginkgolides A, B, and C, bilobalid	Activation of Wnt/β-catenin pathway Expression of VEGF Activation of Akt pathway	[64]
*Centella asiatica*	*Centella asiatica* Linn. Extract	Asiaticosides	Expression of VEGF	[65]

## Data Availability

Data are contained within the article.

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
