# Peer review of "Potential Natural Products Regulation of Molecular Signaling Pathway in Dermal Papilla Stem Cells"

_molecules, 2023, doi:10.3390/molecules28145517_

Round 1

Reviewer 1 Report

This manuscript tried to review the regulation of hair stem cells and the natural product for stem cell stimulating. However, they failed to achieve this. The level of the manuscript is far from the standard of the journal.

1. The structure and the text are not consistent with the title.

2. Some key claims are not accurate. For example, “hair follicle stem cells residing in two areas”.

3. Molecular pathways and natural products for hair follicle stem cells and DP stem cells are usually different. The author discuss them together, which is hard to understand for readers.

4. The figures are mostly a repeat of published references. They should summary to support their conclusion.

5. Table 1 is kind of informative. It is better to summarize the specific functional components of extracts and the origination of the compounds as well. 

Extensive editing of English language required

Author Response

Reviewer 1

This manuscript tried to review the regulation of hair stem cells and the natural product for stem cell stimulating. However, they failed to achieve this. The level of the manuscript is far from the standard of the journal.

  1. The structure and the text are not consistent with the title.

Reply: Thank you very much for your suggestion. We agree and change the title to “Potential natural products regulation for molecular signaling pathway in dermal papilla stem cells.”

  1. Some key claims are not accurate. For example, “hair follicle stem cells residing in two areas”.

Reply: We would like to thank you for your suggestion. We changed it.

  1. Molecular pathways and natural products for hair follicle stem cells and DP stem cells are usually different. The author discuss them together, which is hard to understand for readers.

Reply: Thank you very much for your valuable comment and suggestion. In this review, we want to revise the general effect of natural products on hair growth and how to regulate molecular pathways in hair stem cells, especially DP cell lines. Our primary emphasis was on the DP stem cells; however, we also included a discussion on hair follicle stem cells as an supplementary aspect in our review. We will try to revise it specifically such as hair follicle stem cells and DP stem cells in the next review paper. Now we have changed a little in the section on molecular pathways regulating stem cell properties of hair and highlighted the phrase that we have changed.

  1. The figures are mostly a repeat of published references. They should summary to support their conclusion.

Reply: Thank you very much for your suggestion. We added summarized figure in Manuscript.

  1. Table 1 is kind of informative. It is better to summarize the specific functional components of extracts and the origination of the compounds as well. 

Reply: Thank you very much for your valuable suggestion to improve our review paper. We have modified table 1 according to your comment.

Reviewer 2 Report

The authors prepared an interesting review summarizing research in the field of stem cell stimulation in application to the growth of hair follicles with low molecular weight organic compounds of natural origin, and also presented and discussed the mechanisms of their activation and signaling pathways. I am sure that the presented review will be of interest to a wide audience of Moleculs readers. I believe that, given the level of novelty and originality of the presented review, it allows me to recommend it for publication, after a slight revision.

Recommendations to authors:

- Given the wide audience of the journal, represented mostly by researchers in the field of organic, medicinal chemistry and the chemistry of natural compounds, I believe that the authors of the review should not limit themselves to a verbal description of the chemical compounds contained in various extracts, but give their structural formulas for clarity and improve perception.

Author Response

Reviewer 2

The authors prepared an interesting review summarizing research in the field of stem cell stimulation in application to the growth of hair follicles with low molecular weight organic compounds of natural origin, and also presented and discussed the mechanisms of their activation and signaling pathways. I am sure that the presented review will be of interest to a wide audience of Moleculs readers. I believe that, given the level of novelty and originality of the presented review, it allows me to recommend it for publication, after a slight revision.

Recommendations to authors:

- Given the wide audience of the journal, represented mostly by researchers in the field of organic, medicinal chemistry and the chemistry of natural compounds, I believe that the authors of the review should not limit themselves to a verbal description of the chemical compounds contained in various extracts, but give their structural formulas for clarity and improve perception.

Reply: Thank you for your kindly suggestion on our manuscript to improve the quality. We added updated information such as chemical compounds contained in extracts and their structural formulas in Table 1.

Round 2

Reviewer 1 Report

The manuscript has been revised. However, significant concerns still exist.

1. The basic concept of hair follicle stem cells is confused in the manuscript. It is better to define it in the Introduction and use it consistently.

2 All the structures of the components can easily be found in Google, so they are just a repeat of references. Removing them and summarizing their features in just one table is better.

3. The language needs polishing by a native speaker.

Extensive editing of English language required

Author Response

Response 1

The manuscript has been revised. However, significant concerns still exist.

  1. The basic concept of hair follicle stem cells is confused in the manuscript. It is better to define it in the Introduction and use it consistently.

Response: Thank you very much for your suggestion. In this review manuscript, we emphasis was on the DP stem cells. Therefore, we provide an explanation of the fundamental concept of dermal papilla stem cells in introduction.

2 All the structures of the components can easily be found in Google, so they are just a repeat of references. Removing them and summarizing their features in just one table is better.

Response: Thank you very much for your valuable suggestion. We require additional information regarding the structure and chemical composition of certain natural products obtained from primary sources.

  1. The language needs polishing by a native speaker.

Response: Thank you so much. We have already verified the language using Enago. This one is our supporting evidence.